# BENK: The Beran Estimator with Neural Kernels for Estimating the Heterogeneous Treatment Effect

Stanislav Kirpichenko †, Lev Utkin †, Andrei Konstantinov and Vladimir Muliukha *

Higher School of Artificial Intelligence Technologies, Peter the Great St. Petersburg Polytechnic University, 195251 St. Petersburg, Russia
* Correspondence: vladimir.muliukha@spbstu.ru
† These authors contributed equally to this work.

**Abstract:** A method for estimating the conditional average treatment effect under the condition of censored time-to-event data, called BENK (the Beran Estimator with Neural Kernels), is proposed. The main idea behind the method is to apply the Beran estimator for estimating the survival functions of controls and treatments. Instead of typical kernel functions in the Beran estimator, it is proposed to implement kernels in the form of neural networks of a specific form, called neural kernels. The conditional average treatment effect is estimated by using the survival functions as outcomes of the control and treatment neural networks, which consist of a set of neural kernels with shared parameters. The neural kernels are more flexible and can accurately model a complex location structure of feature vectors. BENK does not require a large dataset for training due to its special way for training networks by means of pairs of examples from the control and treatment groups. The proposed method extends a set of models that estimate the conditional average treatment effect. Various numerical simulation experiments illustrate BENK and compare it with the well-known T-learner, S-learner and X-learner for several types of control and treatment outcome functions based on the Cox models, the random survival forest and the Beran estimator with Gaussian kernels. The code of the proposed algorithms implementing BENK is publicly available.

**Keywords:** treatment effect; survival analysis; Nadaraya–Watson regression; Beran estimator; neural network; meta-learner

## 1. Introduction

Survival analysis is an important and fundamental tool for modeling applications when using time-to-event data [1], which can be encountered in medicine, reliability, safety, finance, etc. This is a reason why many machine learning models have been developed to deal with time-to-event data and to solve the corresponding problems in the framework of survival analysis [2]. The crucial peculiarity of time-to-event data is that a training set consists of censored and uncensored observations. When time-to-event exceeds the duration of an observation, we have a censored observation. When an event is observed, i.e., time-to-event coincides with the duration of the observation, we deal with an uncensored observation.

Many survival models are able to cover various cases of time-to-event probability distributions and their parameters [2]. One of the important models is the Cox proportional hazards model [3], which can be regarded as a semi-parametric regression model. There are also many parametric and nonparametric models. When considering machine learning survival models, it is important to point out that, in contrast to other machine learning models, their outcomes are functions, for instance, survival functions, hazard functions or cumulative hazard functions. For instance, the well-known effective model called the random survival forest (RSF) [4] predicts survival functions (SFs) or cumulative hazard functions.

An important area of survival model application is the problem of treatment effect estimation, which is often solved in the framework of machine learning problems [5]. The treatment effect shows how a treatment may be efficient depending on characteristics of a patient. The problem is solved by dividing patients into two groups called treatment and control, such that patients from the different groups can be compared. One of the popular measures of efficient treatment that is used in machine learning models is the average treatment effect (ATE) [6], which is estimated on the basis of observed data about patients, such as the mean difference between outcomes of patients from the treatment and control groups.

Due to the difference between characteristics of patients and their responses to a particular treatment, the treatment effect is measured using the conditional average treatment effect (CATE), which is defined as the mean difference between outcomes of patients from the treatment and control groups, conditional on a patient feature vector [7]. In fact, most methods of CATE estimation are based on constructing two regression models for controls and treatments. However, two difficulties in CATE estimation can be met. The first one is that the treatment group is usually very small. Therefore, many machine learning models cannot be accurately trained on the small datasets. The second difficulty is fundamental. Each patient cannot be simultaneously in the treatment and control groups, i.e., we either observe the patient outcome under the treatment or control, but never both [8]. Nevertheless, to overcome these difficulties, many methods for estimating CATE have been proposed and developed due to the importance of the problem in many areas [9–13].

One of the approaches for constructing regression models for controls and treatments is the application of the Nadaraya–Watson kernel regression [14,15], which uses standard kernel functions, for instance, the Gaussian, uniform or Epanechnikov kernels. In order to avoid selecting a standard kernel, Konstantinov et al. [16] proposed to implement kernels and the whole Nadaraya–Watson kernel regression by using a set of identical neural subnetworks with shared parameters, with a specific way of the network training. The corresponding method called TNW–CATE (Trainable Nadaraya–Watson regression for CATE) is based on an important assumption that domains of the feature vectors from the treatment and control groups are similar. Indeed, we often treat patients after being in the control group, i.e., it is assumed that treated patients came to the treatment group from the control group. For example, it is difficult to expect that patients with pneumonia will be treated with new drugs for stomach disease. The neural kernels (kernels implemented as the neural network) are more flexible, and they can accurately model a complex location structure of feature vectors, for instance, when the feature vectors from the control and treatment group are located on the spiral, as shown in Figure 1, where small triangular and circle markers correspond to the treatment and control groups, respectively. This is another important peculiarity of the TNW–CATE. Results provided in [16] illustrated outperformance of the TNW–CATE in comparison with other methods when the treatment group was very small and the feature vectors had complex structure.

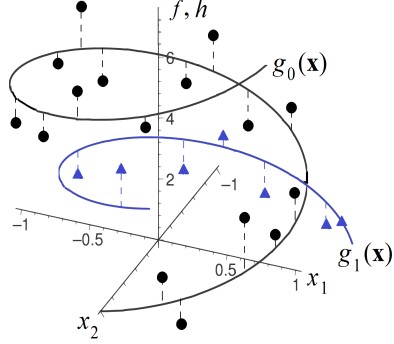

**Figure 1.** An example of the control $g_0(\mathbf{x})$ and treatment $g_1(\mathbf{x})$ functions, which are unknown, and of the control (circle markers) and treatment (triangle markers) data points, which are observed.

Following the ideas behind the TNW–CATE, we propose the CATE estimation method, called BENK (the Beran Estimator with Neural Kernels), dealing with censored time-to-event data in the framework of survival analysis. The main idea behind the proposed method is to apply the Beran estimator [17] for estimating SFs of treatments and controls and to compare them for estimating the CATE. One of the important peculiarities of the Beran estimator is that it takes into account distances between feature vectors by using kernels which measure the similarity between any two feature vectors. On the one hand, the Beran estimator can be regarded as an extension of the Kaplan–Meier estimator. It allows us to obtain SFs that are conditional on the feature vectors, which can be viewed as outcomes of regression survival models for the treatment and control groups. On the other hand, the Beran estimator can also be viewed as an analogue of the Nadaraya–Watson kernel regression for survival analysis. However, typical kernels, for example, the Gaussian one, cannot cope with the possible complex structure of data. Therefore, similarly to the TNW–CATE model, we propose to implement kernels in the Beran estimator by means of neural subnetworks and to estimate CATE by using the obtained SFs. The whole neural network model is trained in an end-to-end manner.

Various numerical experiments illustrate BENK and its peculiarities. They also show that BENK outperforms many well-known meta-models: the T-learner and the S-learner, the X-learner for several control and treatment output functions based on the Cox models, the RSF and the Beran estimator with Gaussian kernels.

BENK is implemented using the framework PyTorch with open code. The code of the proposed algorithms can be found at https://github.com/Stasychbr/BENK (accessed on 27 October 2023).

The paper is organized as follows. Section 2 is a review of the existing CATE estimation models, including CATE estimation survival models, the Nadaraya–Watson regression models and general survival models. A formal statement of the CATE estimation problem is provided in Section 3. The CATE estimation problem in the case of censored data is stated in Section 4. The Beran estimator is considered in Section 5. A description of BENK is provided in Section 6. Numerical experiments illustrating BENK and comparing it with other models can be found in Section 7. Concluding remarks are provided in Section 8.

## 2. Related Work

**Estimating CATE**. One of the important approaches to implement personalized medicine is the treatment effect estimation. As a result, many interesting machine learning models have been developed and implemented to estimate CATE. First, we have to point out an approach which uses the Lasso model for estimating CATE [18]. The SVM was also applied to solve the problem [19]. A unified framework for constructing fast tree-growing procedures for solving the CATE problem was provided in [20]. McFowland et al. [21] estimated CATE by using the anomaly detection model. A set of meta-algorithms or meta-learners, including the T-learner, the S-learner and the X-learner, were studied in [12]. Many other models related to the CATE estimation problem are studied in [22,23].

The aforementioned models are constructed by using machine learning methods, which are different from neural networks. However, neural networks became a basis for developing many interesting and efficient models [24–27].

Due to the importance of the CATE problem, there are many other publications devoted to this problem [28–31].

The next generation of models that solve the CATE estimation problem is based on architectures of transformers with the attention operations [32–34]. The transfer learning technique was successfully applied to the CATE estimation in [35,36]. Ideas of using the Nadaraya–Watson kernel regression in the CATE estimation were studied in [37]. These ideas can lead to the best results under the condition of large numbers of examples in the treatment and control groups. At the same time, a small amount of training data may lead to overfitting and unsatisfactory results. Therefore, the problem of overcoming this possible

limitation motivated researchers to introduce a neural network of a special architecture, which implements the trainable kernels in the Nadaraya–Watson regression [16].

**Machine learning models in survival analysis**. The importance of survival analysis applications can be regarded as one of the reasons for developing many machine learning methods that deal with censored and time-to-event data. A comprehensive review of machine learning survival models is presented in [2]. A large portion of models use the Cox model, which can be viewed as a simple and applicable survival model that establishes a relationship between covariates and outcomes. Various extensions of the Cox model have been proposed. They can be conditionally divided into two groups. The first group remains the linear relationship of covariates and includes various modifications of the Lasso models [38]. The second group of models relaxes the linear relationship assumption accepted in the Cox model [39].

Many survival models are based on using the RSFs, which can be regarded as powerful tools, especially when models learn on tabular data [40,41]. At the same time, there are many survival models based on neural networks [42,43].

**Estimating CATE with censored data**. Censored data can be regarded as an important type, especially for estimating the treatment effect because many applications are characterized by time-to-event data as outcomes. This peculiarity is a reason for developing many CATE models that deal with censored data in the framework of survival analysis [44–46]. Modifications of the survival causal trees and forests for estimating the CATE based on censored observational data were proposed in [44]. An approach combining a treatment-specific semi-parametric Cox loss with a treatment-balanced deep neural network was studied in [47]. Nagpal et al. [48] presented a latent variable approach to model the CATE under assumption that an individual can belong to one of the latent clusters with distinct response characteristics. The problem of CATE estimation by focusing on learning (discrete-time) treatment-specific conditional hazard functions was studied in [49]. A three-stage modular design for estimating CATE in the framework of survival analysis was proposed in [50]. A comprehensive simulation study presenting a wide range of settings, describing CATE by taking into account the covariate overlap, was carried out in [51]. Rytgaard et al. [52] presented a data-adaptive estimation procedure for estimation of the CATE in a time-to-event setting based on generalized random forests. The authors proposed a two-step procedure for estimation, applying inverse probability weighting to construct time-point-specific weighted outcomes as input for the forest. A unified framework for counterfactual inference, applicable to survival outcomes and formulation of a nonparametric hazard ratio metric for evaluating the CATE, were proposed in [53].

In spite of many works and results devoted to estimating the CATE with censored data, these methods are mainly based on assumptions of a large number of examples in the treatment group. Moreover, there are no results implementing the Nadaraya–Watson regression by means of neural networks.

## 3. CATE Estimation Problem Statement

According to the CATE estimation problem, all patients are divided into two groups: control and treatment. Let the control group be the set $\mathcal{C} = \{(\mathbf{x}_1, f_1), \ldots, (\mathbf{x}_c, f_c)\}$ of $c$ patients, such that the $i$-th patient is characterized by the feature vector $\mathbf{x}_i = (x_{i1}, \ldots, x_{id}) \in \mathbb{R}^d$ and the $i$-th observed outcome $f_i \in \mathbb{R}$ (time to event, temperature, the blood pressure, etc.). It is also supposed that the treatment group is the set $\mathcal{T} = \{(\mathbf{y}_1, h_1), \ldots, (\mathbf{y}_t, h_t)\}$ of $t$ patients, such that the $i$-th patient is characterized by the feature vector $\mathbf{y}_i = (y_{i1}, \ldots, y_{id}) \in \mathbb{R}^d$ and the $i$-th observed outcome $h_i \in \mathbb{R}$. The indicator of a group for the $i$-th patient is denoted as $T_i \in \{0, 1\}$, where $T_i = 0$ ($T_i = 1$) corresponds to the control (treatment) group.

We use different notations $\mathbf{x}_i$ and $\mathbf{y}_i$ for controls and treatments in order to avoid additional indices. However, we use the vector $\mathbf{z} \in \mathbb{R}^d$ instead of $\mathbf{x}$ and $\mathbf{y}$ when estimating the CATE.

Suppose that the potential outcomes of patients from the control and treatment groups are $F$ and $H$, respectively. The treatment effect for a new patient with the feature vector $\mathbf{z}$ is

estimated by the individual treatment effect, defined as $H - F$. The fundamental problem of computing the CATE is that only one of the outcomes $f$ or $h$ for each patient can be observed. An important assumption of unconfoundedness [54] is used to allow the untreated patients to be used to construct an unbiased counterfactual for the treatment group [55]. According to the assumption, potential outcomes are characteristics of a patient before the patient is assigned to a treatment condition, or, formally, the treatment assignment $T$ is independent of the potential outcomes for $F$ and $H$ that conditional on the feature vector $\mathbf{z}$, which can be written as

$$T \perp \{F, H\} \mid \mathbf{z}. \tag{1}$$

The second assumption, called the overlap assumption, regards the joint distribution of treatments and covariates. This assumption claims that a positive probability of being both treated and untreated for each value of $\mathbf{z}$ exists. This implies that the following holds with probability 1:

$$0 < \Pr\{T = 1 \mid \mathbf{z}\} < 1. \tag{2}$$

Let $\mathbf{Z}$ be the random feature vector from $\mathbb{R}^d$. The treatment effect is estimated by means of CATE, which is defined as the expected difference between two potential outcomes, as follows [56]:

$$\tau(\mathbf{z}) = \mathbb{E}[H - F \mid \mathbf{Z} = \mathbf{z}]. \tag{3}$$

By using the above assumptions, CATE can be rewritten as

$$\tau(\mathbf{z}) = \mathbb{E}[H \mid \mathbf{Z} = \mathbf{z}] - \mathbb{E}[F \mid \mathbf{Z} = \mathbf{z}]. \tag{4}$$

The motivation behind unconfoundedness is that nearby observations in the feature space can be treated as having come from a randomized experiment [7].

Suppose that functions $g_0(\mathbf{z})$ and $g_1(\mathbf{z})$ express outcomes of the control and treatment patients, respectively. Then, they can be written as follows:

$$f = g_0(\mathbf{z}) + \varepsilon, \ h = g_1(\mathbf{z}) + \varepsilon, \tag{5}$$

where $\varepsilon$ is noise governed by the normal distribution with the zero expectation.

The above imply that the CATE can be estimated as

$$\tau(\mathbf{z}) = g_1(\mathbf{z}) - g_0(\mathbf{z}). \tag{6}$$

An example illustrating the controls (circle markers), treatments (triangle markers) and corresponding unknown function $g_0$ and $g_1$ are shown in Figure 1.

## 4. CATE with Censored Data

Before considering the CATE estimation problem with the censored data, we introduce basic statements of survival analysis. Let us define the training set $D_0$, which consists of $c$ triplets $(\mathbf{x}_i, \delta_i, f_i)$, $i = 1, \ldots, c$, where $\mathbf{x}_i^{\mathrm{T}} = (x_{i1}, \ldots, x_{id})$ is the feature vector characterizing the $i$-th patient from the control group, $f_i$ is the time to the event concerning the $i$-th control patient and $\delta_i \in \{0, 1\}$ is the indicator of censored or uncensored observations. If $\delta_i = 1$, then the event of interest is observed (the uncensored observation). If $\delta_i = 0$, then we have the censored observation. Only the right-censoring is considered when the observed survival time is less than or equal to the true survival time. Many applications of survival analysis deal with the right-censored observations [2]. The main goal of survival machine learning modeling is to use set $D_0$ to estimate probabilistic characteristics of time $F$ to the event of interest for a new patient with the feature vector $\mathbf{z}$.

In the same way, we define the training set $D_1$, which consists of $d$ triplets $(\mathbf{y}_i, \gamma_i, h_i)$, $i = 1, \ldots, s$, where $\mathbf{y}_i^{\mathrm{T}} = (y_{i1}, \ldots, y_{id})$ is the feature vector characterizing the $i$-th patient from the treatment group, $h_i$ is the time to the event concerning the $i$-th treatment patient and $\gamma_i \in \{0, 1\}$ is the indicator of censoring.

The survival function (SF), denoted $S(t \mid \mathbf{z})$, can be regarded as an important concept in survival analysis. It represents the probability of survival of a patient with the feature vector $\mathbf{z}$ up to time $t$, that is, $S(t \mid \mathbf{z}) = \Pr\{T > t \mid \mathbf{z}\}$. The hazard function, denoted $\lambda(t \mid \mathbf{z})$, can be viewed as another concept in survival analysis. It is defined as the rate of an event at time $t$ given that no event occurred before time $t$. It is expressed through the SF as follows:

$$\lambda(t \mid \mathbf{z}) = -\frac{\mathrm{d}}{\mathrm{d}t} \ln S(t \mid \mathbf{z}). \tag{7}$$

The integral of the hazard function, denoted $H(t \mid \mathbf{x})$, is called the cumulative hazard function and can be interpreted as the probability of an event at time $t$ given survival until time $t$, i.e.,

$$\Lambda(t \mid \mathbf{z}) = \int_0^t \lambda(r \mid \mathbf{z}) dr. \tag{8}$$

It is expressed through the SF as follows:

$$\Lambda(t \mid \mathbf{z}) = -\ln(S(t \mid \mathbf{z})). \tag{9}$$

The above functions for controls and treatments are written with indices 0 and 1, respectively, for instance, $S_0(t \mid \mathbf{z}) = \Pr\{F > t \mid \mathbf{z}\}$ and $S_1(t \mid \mathbf{z}) = \Pr\{H > t \mid \mathbf{z}\}$.

In order to compare survival models, Harrell's concordance index, or the C-index [57], is usually used. The C-index measures the probability that, in a randomly selected pair of examples, the example that failed first had a worst predicted outcome. It is calculated as the ratio of the number of pairs correctly ordered by the model to the total number of admissible pairs. A pair is not admissible if the events are both right-censored or if the earliest time in the pair is censored. The corresponding survival model is supposed to be perfect when the C-index is 1. The case when the C-index is 0.5 says that the survival model is the same as random guessing. The case when the C-index is less than 0.5 says that the corresponding model is worse than random guessing.

In contrast to the standard CATE estimation problem statement given in the previous section, the CATE estimation problem with censored data has another statement, which is due to the fact that outcomes in survival analysis are random times to an event of interest having some conditional probability distribution. In other words, predictions corresponding to a patient characterized by vector $\mathbf{z}$ in survival analysis provided by a survival machine learning model are represented in the form of functions of time, for instance, in the form of SF $S(t \mid \mathbf{z})$. This implies that the CATE $\tau(\mathbf{x})$ should be reformulated by taking into account the above peculiarity. It is assumed that SFs as well as hazard functions for control and treatment patients, estimated by using datasets $D_0$ and $D_1$, will have indices 0 and 1, respectively.

The following definitions of the CATE in the case of censored data can be found in [58]:

1.  Difference in expected lifetimes:

$$\tau(\mathbf{z}) = \int_0^{t_{\max}} (S_1(t \mid \mathbf{z}) - S_0(t \mid \mathbf{z})) \mathrm{d}t = \mathbb{E}\{T_1 - T_0 \mid X = \mathbf{z}\}; \tag{10}$$

2.  Difference in SFs:

$$\tau(t, \mathbf{z}) = S_1(t \mid \mathbf{z}) - S_0(t \mid \mathbf{z}); \tag{11}$$

3.  Hazard ratio:

$$\tau(t, \mathbf{z}) = \lambda_1(t \mid \mathbf{z}) / \lambda_0(t \mid \mathbf{z}). \tag{12}$$

We will the first integral definition of the CATE. Let $0 = t_0 < t_1 < \ldots < t_n$ be the distinct times to an event of interest, which are obtained from the set $\{f_1, \ldots, f_c\} \cup \{h_1, \ldots, h_s\}$. The SF provided by a survival machine learning model is a step function, i.e., it can be represented as $S(t \mid \mathbf{z}) = \sum_{j=1}^n S^{(j)}(\mathbf{z}) \cdot \chi_j(t)$, where $\chi_j(t)$ is the indicator

function, taking a value of 1 if $t \in [t_{j-1}, t_j]$; $S^{(j)}(\mathbf{z})$ is the value of the SF in interval $[t_{j-1}, t_j]$. Hence, the following holds:

$$
\begin{aligned}
\tau(\mathbf{z}) &= \int_0^{t_{\max}} (S_1(t \mid \mathbf{z}) - S_0(t \mid \mathbf{z})) \mathrm{d}t \\
&= \sum_{j=1}^n \left( S_1^{(j)}(\mathbf{z}) - S_0^{(j)}(\mathbf{z}) \right) (t_j - t_{j-1}).
\end{aligned}
\tag{13}
$$

## 5. Nonparametric Estimation of Survival Functions and CATE

The idea to use the nonparametric kernel regression for estimating SFs and other concepts of survival analysis has been proposed by several authors [59,60]. One of the interesting estimators is the Beran estimator [17] of the SF, which is defined as follows:

$$
S(t \mid \mathbf{x}) = \prod_{f_i \leq t} \left\{ 1 - \frac{W(\mathbf{x}, \mathbf{x}_i)}{1 - \sum_{j=1}^{i-1} W(\mathbf{x}, \mathbf{x}_j)} \right\}^{\delta_i},
\tag{14}
$$

where $W(\mathbf{x}, \mathbf{x}_i)$ are the kernel weights, defined as

$$
W(\mathbf{x}, \mathbf{x}_i) = \frac{K(\mathbf{x}, \mathbf{x}_i)}{\sum_{j=1}^n K(\mathbf{x}, \mathbf{x}_j)}.
\tag{15}
$$

The above expression is given for the controls. The same estimator can be written for treatments, but $\mathbf{x}$, $\delta_i$, $f_i$ are replaced with $\mathbf{y}$, $\gamma_i$, $h_i$, respectively.

The Beran estimator can be regarded as a generalization of the Kaplan–Meier estimator because the former is reduced to the latter if the kernel weights take values $W(\mathbf{x}, \mathbf{x}_i) = 1/n$. It is also interesting to note that the product in (14) only takes into account uncensored observations, whereas the weights are normalized by using uncensored as well as censored observations.

By using (14) and (13), we can construct a neural network that is trained to implement the weights $W(\mathbf{z}, \mathbf{x}_i)$, $W(\mathbf{z}, \mathbf{y}_i)$ and to estimate SFs $S_1(t \mid \mathbf{z})$ and $S_0(t \mid \mathbf{z})$ for computing $\tau(\mathbf{z})$.

## 6. Neural Network for Estimating CATE

Let us consider how the Beran estimator with neural kernels can be implemented by means of a neural network of a special type. Our first aim is to implement kernels $K(\mathbf{x}, \mathbf{x}_i)$ by means of a neural subnetwork, which is called the neural kernel and is a part of the whole network for implementing the Beran estimator. The second aim is for this network to learn on the control data. Having the trained kernel, we can apply it to compute the conditional survival function for controls, as well as for treatments, because the kernels in (14) do not directly depend on times to events $f_i$ or $h_i$. However, in order to train the kernel, we have to train the whole network because the loss function is defined through SF $S_0(t \mid \mathbf{x})$, which represents the probability of survival of a control patient up to time $t$, which is estimated by means of the Beran estimator. This implies that the whole network contains blocks of the neural kernels for computing kernels $K(\mathbf{x}, \mathbf{x}_i)$, normalization for computing the kernel weights $W(\mathbf{x}, \mathbf{x}_i)$ and the Beran estimator in accordance with (14). In order to realize a training procedure for the network, we randomly select a portion ($n$ examples) from all control training examples and form a single specific example from $n$ selected ones. This random selection is repeated $N$ times to have $N$ examples for training. Thus, for every $\mathbf{x}_i$, $i = 1, \ldots, c$, from the control group, we add another vector $\mathbf{x}_k$ from the same set of controls. By composing $n$ pairs of vectors $(\mathbf{x}_i, \mathbf{x}_k)$, $k = 1, \ldots, n$, and including other elements of training examples $(\delta_i, f_i)$, we obtain one composite vector of data, representing one new training example for the entire neural network. Such new training examples can be constructed for each $i = 1, \ldots, c$. The formal construction of the training set is considered below.

Having the trained neural kernel, it can be successfully used for computing SF $S_0(t \mid \mathbf{z})$ of controls and SF $S_1(t \mid \mathbf{z})$ of treatments for arbitrary vectors of features $\mathbf{z}$, again applying the Beran estimator.

Let us consider the training algorithm in detail. First, we return to the set of $c$ controls $\mathcal{C} = \{(\mathbf{x}_i, \delta_i, f_i), \ i = 1, \ldots, c\}$. For every $i$ from set $\{1, \ldots, c\}$, we construct $N$ subsets $\mathcal{C}_i^{(r)}$, $r = 1, \ldots, N$, having $n$ examples randomly selected from $\mathcal{C} \backslash (\mathbf{x}_i, \delta_i, f_i)$, which have indices from the index set $\mathcal{I}^{(r)}$, i.e., the subsets $\mathcal{C}_i^{(r)}$ are of the form

$$\mathcal{C}_i^{(r)} = \{(\mathbf{x}_k^{(r)}, \delta_k^{(r)}, f_k^{(r)}), \ k \in \mathcal{I}^{(r)}\}, \ r = 1, \ldots, N. \tag{16}$$

Here, $N$ and $n$ can be regarded as tuning hyperparameters. Upper index $r$ indicates that the $r$-th example $(\mathbf{x}_k^{(r)}, \delta_k^{(r)}, f_k^{(r)})$ is randomly taken from $\mathcal{C} \backslash (\mathbf{x}_i, \delta_i, f_i)$, i.e., there is an example $(\mathbf{x}_j, \delta_j, f_j)$ from $\mathcal{C}$ such that $\mathbf{x}_k^{(r)} = \mathbf{x}_j$, $\delta_k^{(r)} = \delta_j$, $f_k^{(r)} = f_j$. Each subset $\mathcal{C}_i^{(r)}$, jointly with $(\mathbf{x}_i, \delta_i, f_i)$, forms a training example $\mathbf{a}_i^{(r)}$ for the control network as follows:

$$\mathbf{a}_i^{(r)} = \left( \mathcal{C}_i^{(r)}, \mathbf{x}_i, \delta_i, f_i \right), \ i = 1, \ldots, c, \ r = 1, \ldots, N. \tag{17}$$

The number of possible examples $\mathbf{a}_i^{(r)}$ is $c \cdot N$, and these examples are used for training the neural network, whose output is the estimate of SF $\tilde{S}_0(t \mid \mathbf{x}_i)$.

The architecture of the neural network, consisting of $n$ subnetworks that implement the neural kernels, is shown in Figure 2. Examples $\mathbf{a}_i^{(r)}$ produced from the dataset of controls are fed to the whole neural network, such that each pair $(\mathbf{x}_i, \mathbf{x}_k^{(r)})$, $k \in \mathcal{I}^{(r)}$, is fed to each subnetwork, which implements the kernel function. The output of each subnetwork is kernel $K(\mathbf{x}_i, \mathbf{x}_k^{(r)})$. All subnetworks are identical and have shared weights. After normalizing the kernels, we obtain $n$ weights $W(\mathbf{x}_i, \mathbf{x}_k^{(r)})$, which are used to estimate SFs by means of the Beran estimator in (14). The block of the whole neural network that implements the Beran estimator uses all weights $W(\mathbf{x}_i, \mathbf{x}_k^{(r)})$, $k \in \mathcal{I}^{(r)}$, and the corresponding values $\delta_k^{(r)}$ and $f_k^{(r)}$, $k \in \mathcal{I}^{(r)}$. As a result, we obtain SF $\tilde{S}_0(t \mid \mathbf{x}_i)$. In the same way, we compute SFs $\tilde{S}_0(t \mid \mathbf{x}_k)$ for all $k = 1, \ldots, c$. These functions are the basis for training. In fact, the normalization block and the block that implements the Beran estimator can be regarded as part of the neural network, and they are trained in an end-to-end manner.

According to (13), expected lifetimes are used to compute the CATE $\tau(\mathbf{z})$. Therefore, the whole network is trained by means of the following loss function:

$$L = \frac{1}{c^* \cdot N} \sum_{i \in \mathcal{C}^*} \sum_{k=1}^{N} \left( \tilde{E}_k^{(i)} - f_k^{(i)} \right)^2. \tag{18}$$

Here, $\mathcal{C}^*$ is a subset of $\mathcal{C}$, which contains only uncensored examples from $\mathcal{C}$, $c^*$ is the number of elements in $\mathcal{C}^*$; $f_k^{(i)}$ is the time to an event of the $k$-th example from the set $\mathcal{C}^* \backslash (\mathbf{x}_i, \delta_i, f_i)$ and $\tilde{E}_k^{(i)}$ is the expected lifetime computed through SF $\tilde{S}_0(t \mid \mathbf{x}_k)$, obtained by integrating the SF:

$$\tilde{E}_k^{(i)} = \sum_{j=1}^{n} (f_j^{(i)} - f_{j-1}^{(i)}) \tilde{S}_0(f_j^{(i)} \mid \mathbf{x}_k). \tag{19}$$

The sum in (18) is taken over uncensored examples from $\mathcal{C}$. However, the Beran estimator uses all the examples.

One of the loss functions, which takes into account all data (censored and uncensored), is the C-index. However, our aim is not to estimate the SF or the CHF. We aim to estimate the difference between the predicted time to event and the expected time to event. Therefore, we use the standard mean squared error (MSE) loss function. But the censored times introduce bias into MSE and, therefore, they are not used.

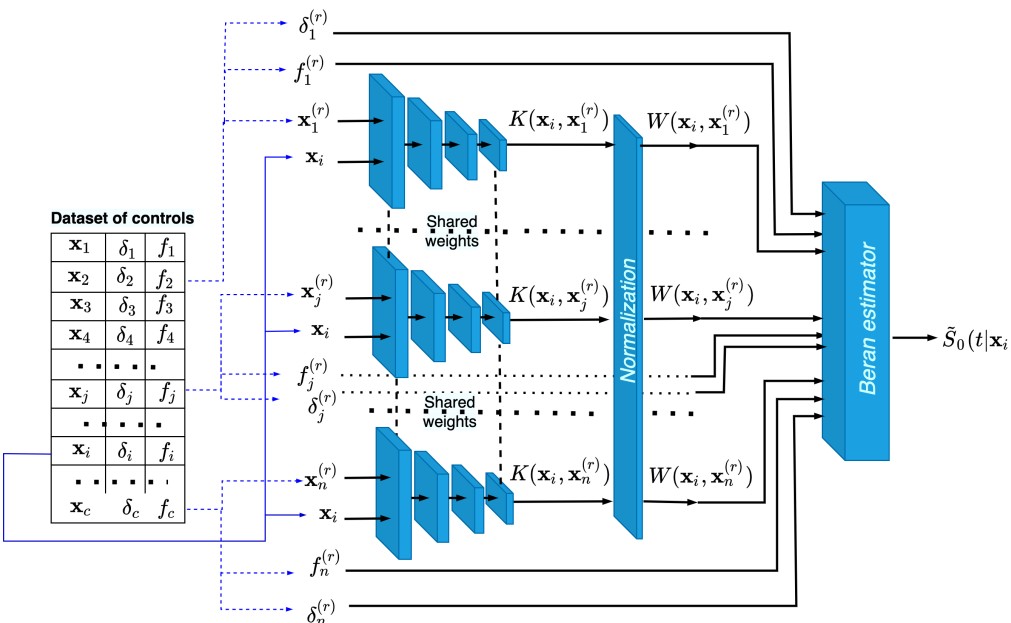

**Figure 2.** The neural network training on examples $\mathbf{a}_i^{(r)}$, composed of controls, for producing the Beran estimator in the form of SF $\tilde{S}_0(t \mid \mathbf{x}_i)$.

It is important to point out that our aim is to train subnetworks with shared training parameters, which are the neural kernels. By having the trained neural kernels, we can use them to compute kernels $K(\mathbf{z}, \mathbf{x}_i)$ and $K(\mathbf{z}, \mathbf{y}_i)$ and then to compute estimates of SFs $\tilde{S}_0(t \mid \mathbf{z})$ and $\tilde{S}_1(t \mid \mathbf{z})$ for controls and treatments, respectively, i.e., we realize the idea of transferring tasks from the control group to the treatment group. Let $t_1^{(0)} < t_2^{(0)} < \ldots < t_c^{(0)}$ and $t_1^{(1)} < t_2^{(1)} < \ldots < t_s^{(1)}$ be the ordered time moments corresponding to times $f_1, \ldots, f_c$ and $h_1, \ldots, h_s$, respectively. Then, the CATE $\tau(\mathbf{z})$ can be computed through SFs $S_1(t \mid \mathbf{z})$ and $S_0(t \mid \mathbf{z})$, again by using the Beran estimators with the trained neural kernels, i.e., in accordance with (13), it holds that

$$\tau(\mathbf{z}) = \sum_{j=1}^{s}(t_j^{(1)} - t_{j-1}^{(1)})\tilde{S}_1^{(j)}(\mathbf{z}) - \sum_{k=1}^{c}(t_k^{(0)} - t_{k-1}^{(0)})\tilde{S}_0^{(k)}(\mathbf{z}), \tag{20}$$

where $\tilde{S}_1^{(j)}(\mathbf{z})$ is the estimation of the SF of treatments on the interval $[t_{j-1}^{(1)}, t_j^{(1)})$, $\tilde{S}_0^{(k)}(\mathbf{z})$ is the estimation of SF of controls in interval $[t_{k-1}^{(0)}, t_k^{(0)})$ and it is assumed that $t_0^{(0)} = t_0^{(1)} = 0$.

The illustration of the neural networks that predict $K(\mathbf{z}, \mathbf{x}_i)$ and $K(\mathbf{z}, \mathbf{y}_i)$ for a new vector $\mathbf{z}$ of features is shown in Figure 3. It can be seen from Figure 3 that the first neural network consists of $c$ subnetworks, such that pairs of vectors $(\mathbf{z}, \mathbf{x}_i)$, $i = 1, \ldots, c$, are fed to the subnetworks, where $\mathbf{x}_i$ is taken from the dataset of controls. Predictions of the first neural network are $c$ kernels $K(\mathbf{z}, \mathbf{x}_i)$, which are used to compute $\tilde{S}_0(t \mid \mathbf{z})$ by means of the Beran estimator (14). The same architecture has the neural network for predicting kernels $K(\mathbf{z}, \mathbf{y}_i)$, used for estimating the treatment SF $\tilde{S}_1(t \mid \mathbf{z})$. This network consists of $s$ subnetworks and uses vectors $\mathbf{y}_i$ from the dataset of treatments. After computing estimates $\tilde{S}_0(t \mid \mathbf{z})$ and $\tilde{S}_1(t \mid \mathbf{z})$, we can find the CATE $\tau(\mathbf{z})$.

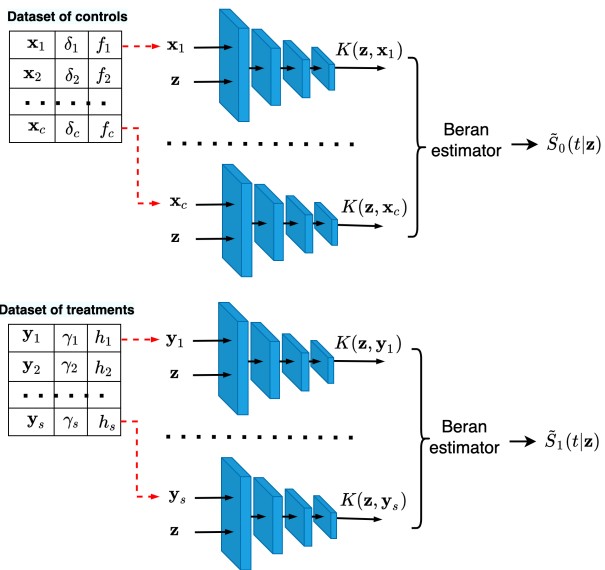

**Figure 3.** Neural networks consisting of the $c$ and $s$ trained neural kernels, predicting new values of kernels $K(\mathbf{z}, \mathbf{x}_i)$ and $K(\mathbf{z}, \mathbf{y}_i)$ that correspond to controls and treatments for computing estimates of $S_1(t \mid \mathbf{z})$ and $S_0(t \mid \mathbf{z})$, respectively.

Phases of training and computing CATE $\tau(\mathbf{x})$ by means of neural kernels are schematically shown as Algorithms 1 and 2, respectively.

---

**Algorithm 1** The algorithm for training neural kernels

---

**Require:** Datasets $\mathcal{C}$ of $c$ controls and $\mathcal{T}$ of $s$ treatments, number $N$ of generated subsets $\mathcal{C}_i^{(r)}$ of $\mathcal{C}$, number of examples in generated subsets $n$

**Ensure:** Neural kernels $K(\cdot, \cdot)$ for their use in the Beran estimator for control and treatment data

1: **for** $i = 1, i \le c$ **do**
2:   **for** $r = 1, r \le N$ **do**
3:    Generate subset $\mathcal{C}_i^{(r)} \subset \mathcal{C} \backslash (\mathbf{x}_i, y_i)$
4:    Form example $\mathbf{a}_i^{(r)} = \left( \mathcal{C}_i^{(r)}, \mathbf{x}_i, \delta_i, f_i \right)$
5:   **end for**
6: **end for**
7: Train the weight sharing neural network with the loss function given in (18) on the set of examples $\mathbf{a}_i^{(r)}$

---

**Algorithm 2** The algorithm for computing CATE for a new feature vector $\mathbf{z}$

---

**Require:** Trained neural kernels, datasets $\mathcal{C}$ and $\mathcal{T}$, testing example $\mathbf{z}$

**Ensure:** CATE $\tau(\mathbf{x})$

1: **for** $i = 1, i \le c$ **do**
2:   Form pair $(\mathbf{z}, \mathbf{x}_i)$ of vectors by using the dataset $\mathcal{C}$ of controls
3:   Feed pair $(\mathbf{z}, \mathbf{x}_i)$ to the trained neural kernel and predict $K(\mathbf{z}, \mathbf{x}_i)$
4: **end for**
5: **for** $i = 1, i \le s$ **do**
6:   Form pair $(\mathbf{z}, \mathbf{y}_i)$ of vectors by using the dataset $\mathcal{T}$ of treatments
7:   Feed pair $(\mathbf{z}, \mathbf{y}_i)$ to the trained neural kernel and predict $K(\mathbf{z}, \mathbf{y}_i)$
8: **end for**
9: Compute $W(\mathbf{z}, \mathbf{x}_i), i = 1, \ldots, c$, $W(\mathbf{z}, \mathbf{y}_i), i = 1, \ldots, s$
10: Estimate $\tilde{S}_0(t \mid \mathbf{x}_k)$ and $\tilde{S}_1(t \mid \mathbf{y}_k)$ using (14)
11: Compute $\tau(\mathbf{x})$ using (20)

## 7. Numerical Experiments

Numerical experiments for studying BENK and its comparison with available models are performed by using simulated datasets because the true CATEs are unknown due to the fundamental problem of causal inference for real data [8]. This implies that control and treatment datasets are randomly generated in accordance with predefined outcome functions.

### 7.1. CATE Estimators for Comparison and Their Parameters

For investigating BENK and its comparison, we use nine models, which can be united in three groups (the T-learner, the S-learner, the X-learner), such that each group is based on three base models for estimating SFs (the RSF, the Cox model, the Beran estimator with Gaussian kernels). The models are given below in terms of survival models:

1. The T-learner [12] is a model which estimates the control SF $S_0(t \mid \mathbf{z})$ and the treatment SF $S_1(t \mid \mathbf{z})$ for every $\mathbf{z}$. The CATE in this case is defined in accordance with (13);
2. The S-learner [12] is a model which estimates SF $S(t \mid \mathbf{z}, T)$ instead of $S_0(t \mid \mathbf{z})$ and $S_1(t \mid \mathbf{z})$, where the treatment assignment indicator $T_i \in \{0, 1\}$ is included as an additional feature to the feature vector $\mathbf{z}_i$. As a result, we have a modified dataset

$$\mathcal{D} = \{(\mathbf{z}_1^*, \delta_1, f_1), \ldots, (\mathbf{z}_c^*, \delta_c, f_c), (\mathbf{z}_{c+1}^*, \gamma_1, h_1), \ldots, (\mathbf{z}_{c+s}^*, \gamma_s, h_s)\}, \tag{21}$$

where $\mathbf{z}_i^* = (\mathbf{x}_i, T_i) \in \mathbb{R}^{d+1}$ if $T_i = 0$, $i = 1, \ldots, c$, and $\mathbf{z}_{c+i}^* = (\mathbf{y}_i, T_i) \in \mathbb{R}^{d+1}$ if $T_i = 1$, $i = 1, \ldots, t$. The CATE is determined as

$$\tau(\mathbf{z}) = \sum_{j=1}^{s} (t_j^{(1)} - t_{j-1}^{(1)}) \tilde{S}^{(j)}(\mathbf{z}, 1) - \sum_{k=1}^{c} (t_k^{(0)} - t_{k-1}^{(0)}) \tilde{S}^{(k)}(\mathbf{z}, 0); \tag{22}$$

3. The X-learner [12] is based on computing the so-called imputed treatment effects and is represented in the following three steps. First, the outcome functions $g_0(\mathbf{x})$ and $g_1(\mathbf{y})$ are estimated using a regression algorithm. Second, the imputed treatment effects are computed as follows:

$$D_1(\mathbf{y}_i) = h_i - g_0(\mathbf{y}_i), \ D_0(\mathbf{x}_i) = g_1(\mathbf{x}_i) - f_i. \tag{23}$$

Third, two regression functions $\tau_1(\mathbf{y})$ and $\tau_0(\mathbf{x})$ are estimated for imputed treatment effects $D_1(\mathbf{y})$ and $D_0(\mathbf{x})$, respectively. The CATE for a point $\mathbf{z}$ is defined as a weighted linear combination of the functions $\tau_1(\mathbf{z})$ and $\tau_0(\mathbf{z})$ as $\tau(\mathbf{z}) = \alpha \tau_0(\mathbf{z}) + (1 - \alpha) \tau_1(\mathbf{z})$, where $\alpha \in [0, 1]$ is a weight that is equal to the ratio of treated patients. The original X-learner does not deal with censored data. Therefore, we propose a simple survival modification of the X-learner. It is assumed that $g_0(\mathbf{y}_i)$ and $g_1(\mathbf{x}_i)$ are expectations $E_0(\mathbf{y}_i)$ and $E_1(\mathbf{x}_i)$ of the times to an event corresponding to control and treatment data, respectively. Expectations $E_0(\mathbf{y}_i)$ and $E_1(\mathbf{x}_i)$ are computed by means of one of the algorithms for determining estimates of SFs $S_0(t \mid \mathbf{z})$ and $S_1(t \mid \mathbf{z})$. The functions $\tau_1(\mathbf{y})$ and $\tau_0(\mathbf{x})$ are implemented using the random forest regression algorithm for all the basic models.

Estimations of SFs $S_0(t \mid \mathbf{z})$ and $S_1(t \mid \mathbf{z})$ as well as $S(t \mid \mathbf{z}, T)$ are carried out by means o the following survival regression algorithms:

1. The RSF parameters of random forests used in experiments are the following:

   - The numbers of trees are 10, 50, 100, 200;
   - The depths are 3, 4, 5, 6;
   - The smallest values of examples which fall in a leaf are 1 example, 1%, 5%, 10% of the training set.

   The above values for the hyperparameters are tested, choosing those leading to the best results;

2. The Cox proportional hazards model [3], which is used with the elastic net regularization with the 3 to 1 ratio coefficient $L_1/L_2$;
3. In contrast to the proposed BENK model, we use the Beran estimator with the standard Gaussian kernels. Values $10^i$, $i = -4, \ldots, 3$, and also values 0.5, 5, 50, 200, 500, 700 of the bandwidth parameter of the Gaussian kernel are tested, choosing those leading to the best results.

In sum, we have nine models for comparison, whose notations are given in Table 1.

**Table 1.** Notations of the models, depending on meta-learners and base models.

| | **Meta-Model** | | |
|---|---|---|---|
| Survival regression algorithms | T-learner | S-learner | X-learner |
| Beran estimator | T-Beran | S-Beran | X-Beran |
| Cox model | T-Cox | S-Cox | X-Cox |
| RSF | T-SF | S-SF | X-SF |

*7.2. Generating Synthetic Datasets*

As has been described above, we consider generating the artificial complex feature spaces and outcomes in the numerical experiments. All the vectors of features, including controls **x** and treatments **y**, are generated by means of three functions: the spiral function, the bell-shaped function and the circular function. The idea to use these functions stems from the goal to obtain complex structures of data, which are poorly processed by many standard methods. The above functions are defined through a parameter $\xi$ as follows:

1. Spiral functions: The feature vectors, having dimensionality $d$ and being located on the Archimedean spirals, are defined for even $d$ as

$$\mathbf{x} = (\xi \sin(\xi), \xi \cos(\xi), \ldots, \xi \sin(\xi \cdot d/2), \xi \cos(\xi \cdot d/2)), \tag{24}$$

and for odd $d$ as

$$\mathbf{x} = (\xi \sin(\xi), \xi \cos(\xi), \ldots, \xi \sin(\xi \cdot \lceil d/2 \rceil)). \tag{25}$$

Values of $\xi$ are uniformly generated from the interval $[0, 10]$ for all numerical experiments;

2. Bell-shaped functions: Features are represented as a set of almost non-overlapping Gaussians. As $\xi$ is uniformly generated in the numerical experiments, we can define $\xi_{\min}$ and $\xi_{\max}$ as corresponding bounds of the uniform distribution. Therefore, the feature vector of dimensionality $d$ is represented as

$$\mathbf{x} = (x_0, x_1, \ldots, x_{d-1}),$$
$$\sigma = \frac{\xi_{\max} - \xi_{\min}}{6d}, \ \mu = \frac{\xi_{\max} - \xi_{\min}}{d-1},$$
$$x_i = \frac{1}{\sigma\sqrt{2\pi}} \cdot \exp\left(\frac{-(\xi - i \cdot \mu)^2}{2\sigma^2}\right), \ i = 1, \ldots, d-1. \tag{26}$$

Therefore, each feature $x_i$ corresponds to its own region in the $\xi$ distribution;

3. Circular functions: The corresponding feature space is generated by using only the even numbers of features. The feature vectors are located on the two-dimensional circles as follows:

$$c_{num} = \frac{d}{2}, \; c_{range} = \frac{\xi_{\max} - \xi_{\min}}{c_{num}},$$
$$\mathbf{x} = (x_1^1, x_1^2, x_2^1, x_2^2, \ldots, x_{c_{num}}^1, x_{c_{num}}^2),$$
$$x_i^1 = \sin\left(\frac{2\pi(\xi - (i-1) \cdot c_{range})}{c_{range}}\right) \cdot \mathbf{I}_i,$$
$$x_i^2 = \cos\left(\frac{2\pi(\xi - (i-1) \cdot c_{range})}{c_{range}}\right) \cdot \mathbf{I}_i,$$
$$\mathbf{I}_i = \mathbf{I}\{(i-1) \cdot c_{range} \le \xi < i \cdot c_{range}\}, \; i = 1, \ldots, c_{num}, \tag{27}$$

where $\mathbf{I}_i$ is an indicator function.

Each pair of features $(x_i^{(1)}, x_i^{(2)})$ corresponds to their own two-dimensional circle and to their own region in the $\xi$ distribution.

In all experiments, feature vectors $\mathbf{y}$ are generated in the same way as vectors $\mathbf{x}$. However, for feature vectors $\mathbf{x}$ and $\mathbf{y}$, from the control and treatment groups, the corresponding times to events $f$ and $h$ are different and are generated by using the Weibull distribution, as follows:

$$f(\xi) = -\left(\frac{\log(u)}{0.0005 \cdot \exp(1.6 \cdot \xi)}\right)^{1/2}, \tag{28}$$

$$h(\xi) = -\left(\frac{\log(u)}{0.005 \cdot \exp(0.8 \cdot \xi)}\right)^{1/2}, \tag{29}$$

where $u$ is the random variable, uniformly distributed on the interval $(0, 1)$; values $f$ and $h$ larger than 2000 are clipped to this value.

This way for generating $f$ and $h$ is in agreement with the Cox model. Hence, we can use the Cox model as a base model among RSFs and the Beran estimator with Gaussian kernels in the numerical experiments.

The proportion of censored data, denoted as $p$, is taken as 33% of all observations in the experiments. Hence, parameters of censoring $\delta_i$ and $\gamma_i$ are generated from the binomial distribution with probabilities $\Pr\{\delta_i = 1\} = \Pr\{\gamma_i = 1\} = 0.67$, $\Pr\{\delta_i = 0\} = \Pr\{\gamma_i = 0\} = 0.33$.

The Precision in Estimation of Heterogeneous Effects metric (PEHE), proposed in [61], is used to reduce the variance in the numerical experiments. According to [61], this metric evaluates the ability of each method to capture treatment effect heterogeneity.

If we label the test dataset as $\mathcal{Z}$, then the PEHE can be defined as follows:

$$\text{PEHE}(\mathcal{Z}) = \sqrt{\frac{1}{N_z} \sum_{\mathbf{z} \in \mathcal{Z}} (\mathbb{E}[(h - f) \mid \mathbf{z}(\xi)] - \tau(\mathbf{z}))^2},$$
$$\mathbb{E}(f \mid \mathbf{z}(\xi)) = \frac{1}{\sqrt{0.0005 \cdot \exp(1.6 \cdot \xi)}} \Gamma\left(\frac{3}{2}\right),$$
$$\mathbb{E}(h \mid \mathbf{z}(\xi)) = \frac{1}{\sqrt{0.005 \cdot \exp(0.8 \cdot \xi)}} \Gamma\left(\frac{3}{2}\right), \tag{30}$$

where $N_z$ is the size of the set $\mathcal{Z}$, taken for all numerical experiments as $N_z = 1000$.

The proportion of treatments and controls in most experiments is 20%, except for experiments studying how the proportion of treatments impacts the CATE, where the proportion of treatments and controls is denoted as $q$. For example, if 100 controls are generated for an experiment with $q = 0.2$, then 20 treatments are generated in addition to controls, such that the total number of examples is 120. The generated feature vectors in all experiments consist of 10 features; the volume of the $\mathcal{C}$ set is 300 unless otherwise stated.

To select optimal hyperparameters of BENK, additional validation examples are generated, such that they belong to only the control group, and the size of this additional validation set is 50% of the set $\mathcal{C}$ size. After the BENK neural network training, this validation set is concatenated with $\mathcal{C}$ for other models, which are trained using cross-validation with three splits. For studying the dependencies, we repeat the numerical experiments 100 times and provide the mean values across these 100 iterations.

Each subnetwork is a fully connected neural network consisting of five layers, with corresponding activation functions ReLU6, ReLU6, ReLU6, Tanh, Softplus. Inputs for each subnetwork are represented in the form $\left\| \mathbf{x}_i - \mathbf{x}_j \right\|$ to ensure the symmetry property of kernels. The non-negativity property of neural kernels is achieved by using the activation function Softplus in the last layer of the subnetworks, which ensures that the output is always positive.

### 7.3. Study of the BENK Properties

In all pictures illustrating results of numerical experiments, dotted curves correspond to the T-learner (triangle markers), the S-learner (triangle markers) or the X-learner (the circle marker) under the condition of using the Beran estimator with the Gaussian kernels. Dash-and-dot curves correspond to the Cox models. Dashed curves with the same markers correspond to the same models implemented using RSFs. The solid curve with cross markers corresponds to BENK. The PEHE metric is used to represent results of experiments. The smaller the values of the PEHE, the better the obtained results. To avoid clutter of curves on the figures, we pick the best model for each T-,S- or X-learner obtained in each experiment.

First, we study different CATE estimators using different numbers $c$ of controls, taking the values 100, 200, 300, 500, 1000. The number of treatments $t$ is determined as 20% of the number of controls. Values of $n$ are equal to $\min\{t, 100\}$. Figures 4–6 illustrate how values of the PEHE metric depend on the number $c$ of controls for different estimators when different functions are used for generating examples. Figure 4 shows the difference between the PEHE metric of BENK and other models in the experiment, with the feature vectors located around the spiral. The T-SF, S-Beran and X-SF models are provided in Figure 4 because they show the best competitive metric values. In order to illustrate how the variance in results depends on the amount of input data, the error bars are also depicted in Figure 4. It can be seen from Figure 4 that the variance in results is reduced with the number of controls. This property of results indicates that the neural network is properly trained. We do not add the error bars to other graphs so as to not mask the relative positions of the corresponding curves. Figure 5 illustrates similar dependencies when the bell-shaped function is used for generating the feature vectors. The selected models in this case are T-Cox, S-SF and X-Cox. Figure 6 illustrates the relationship between different models obtained on the circular feature space. The competitive algorithms given in the picture are T-Beran, S-Beran and X-Beran. It can be seen from Figures 4–6 that the proposed model BENK provides better results in comparison with other models. The largest relative difference between BENK and other models can be observed when the feature vectors are generated in accordance with the spiral function. This function produces the most complex data structure, such that other studied models cannot cope with it.

Another interesting question is how the CATE estimators depend on the proportion $q$ of treatments and controls in the training set. Particularly, for the proposed BENK model, we try to study whether an increasing number of treatments (the set $\mathcal{T}$) provides better CATE results with an unchanged number of controls (the set $\mathcal{C}$). The corresponding numerical results are shown in Figures 7–9. One can see from Figures 7–9 that the enhancement in the PEHE is sufficient in comparison with other CATE estimators when $q$ is changed from 10% to 20% in the experiments with the spiral and bell-shaped functions. Moreover, we again observe the outperformance of BENK in comparison with other estimators.

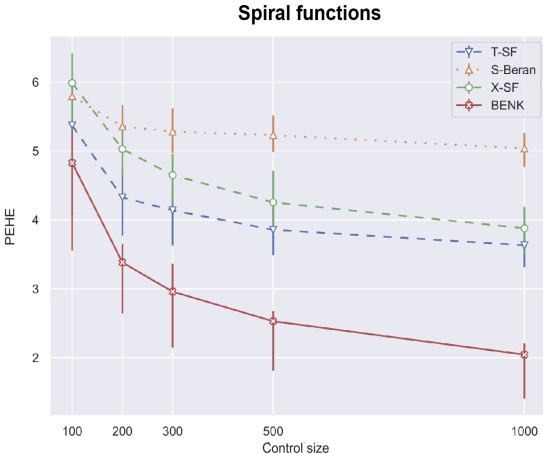

**Figure 4.** The PEHE metric as a function of the number of the controls when the spiral function is used for generating examples.

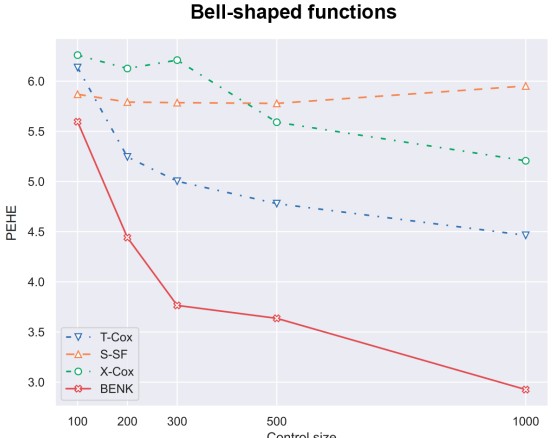

**Figure 5.** The PEHE metric as a function of the number of controls when the bell-shaped function is used for generating examples.

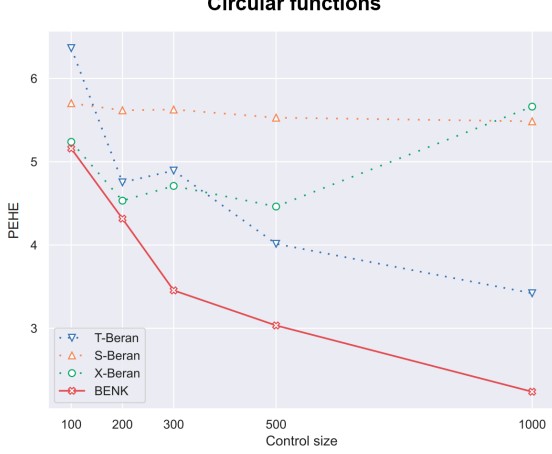

**Figure 6.** The PEHE metric as a function of the number of controls when the circular function is used for generating examples.

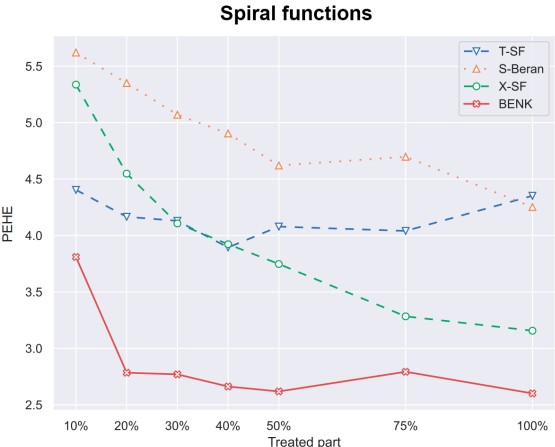

**Figure 7.** The PEHE metric as a function of the part of treatments when the spiral function is used for generating examples.

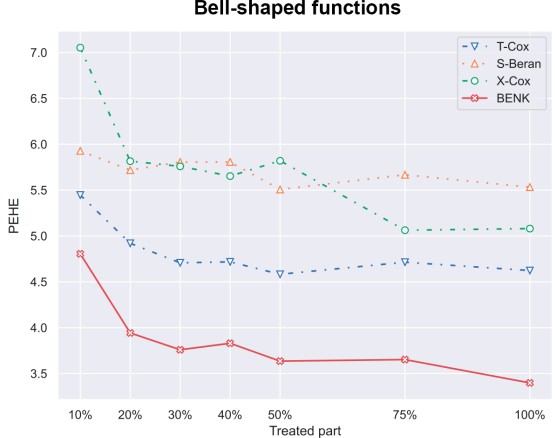

**Figure 8.** The PEHE metric as a function of the part of treatments when the bell-shaped function is used for generating examples.

In the previous experiments, the amount of the censored data was taken $p = 33\%$ of all observations. However, it is interesting to study how this amount impacts the PEHE of the CATE estimators. Figures 10–12 illustrate the corresponding dependencies when different generating functions are used. It can be seen from Figures 10–12 that the PEHE metrics for all estimators, including BENK, increase with the amount of censored data.

Table 2 aims to quantitatively compare results under the following conditions: $c = 400$, $s = 40$, $p = 0.2$, $m = 20$, $N = 1000$. One can see from Table 2 that BENK provides outperforming results. Let us compare results obtained for BENK with the results provided by other models in Table 2. For comparison, we can apply the standard t-test. The obtained $p$-values for all pairs of models are shown in the last column. We can see from Table 2 that all $p$-values are smaller than 0.05. Hence, we can conclude that the outperformance of BENK is statistically significant. It is interesting to note from Table 2 that methods based on the Cox model (T-Cox, S-Cox, X-Cox) show worse results. This can be explained by the weak assumption of the linear relationship of features, which takes place in the Cox model. This assumption contradicts the complex spiral, bell-shaped and circular functions and does not allow us to obtain better results. It should be pointed out that T-NW provides the best result for the bell-shaped generating function among results given by methods other than BENK. This is explained by the fact that the bell-shaped function is close to the Gaussian function; therefore, the method based on using Nadaraya–Watson kernel regression does not crucially differ from BENK. It is also interesting to note that the efficient

methods such as the S-learner and the X-learner often provide worse results in comparison with the T-learner, which is rather weak in standard CATE tasks. This is due to peculiarities of survival data, which differ from the standard regression and classification data.

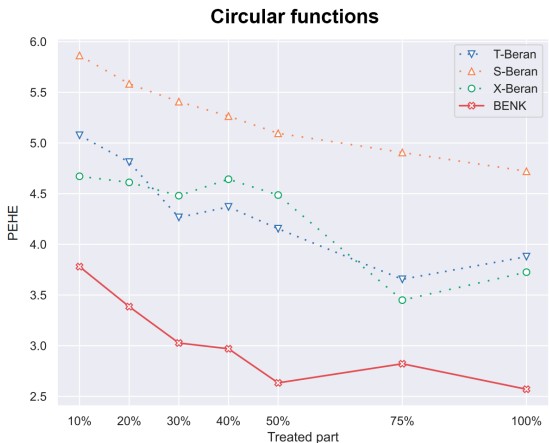

**Figure 9.** The PEHE metric as a function of the part of treatments when the circular function is used for generating examples.

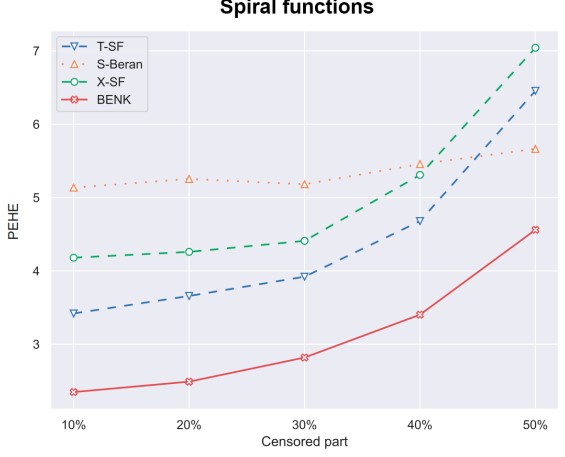

**Figure 10.** The PEHE metric as a function of the amount of censored observations in the training dataset when the spiral function is used for generating examples.

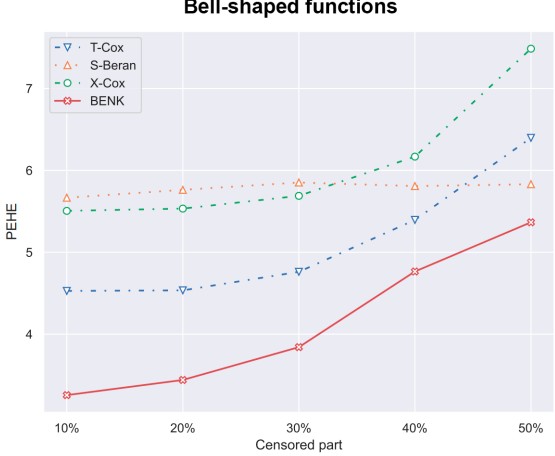

**Figure 11.** The PEHE metric as a function of the amount of censored observations in the training dataset when the bell-shaped function is used for generating examples.

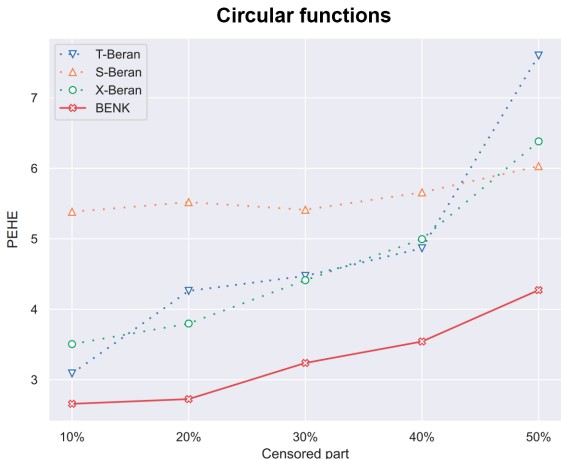

**Figure 12.** The PEHE metric as a function of the amount of censored observations in the training dataset when the circular function is used for generating examples.

**Table 2.** The PEHE values of CATE for different models obtained via different generating functions and the corresponding *p*-values.

| | Generating Functions | | | |
|---|---|---|---|---|
| **Model** | **Spiral** | **Bell-Shaped** | **Circular** | ***p*-Value** |
| T-NW | 5.876 | 4.868 | 5.713 | 0.0457 |
| S-NW | 5.759 | 5.868 | 5.946 | 0.0121 |
| X-NW | 4.985 | 5.090 | 6.317 | 0.0149 |
| T-Cox | 6.198 | 6.518 | 6.126 | 0.0128 |
| S-Cox | 5.959 | 5.941 | 5.963 | 0.0112 |
| X-Cox | 6.331 | 7.396 | 8.357 | 0.0178 |
| T-SF | 3.721 | 5.563 | 6.460 | 0.0401 |
| S-SF | 5.959 | 5.900 | 5.882 | 0.0035 |
| X-SF | 4.853 | 6.339 | 7.176 | 0.0154 |
| BENK | 2.373 | 3.288 | 3.570 | |

It should be noted that we did not provide results of various deep neural network extensions of the CATE estimators because they have not been successful. The problem is that neural networks require a large amount of data for training and the considered small datasets have led to overfitting the networks. This is why we studied models which provide satisfactory predictions under condition of small amounts of data.

## 8. Conclusions

A new method called BENK for solving the CATE problem under the condition of censored data has been presented. It extends the idea behind TNW–CATE proposed in [16] to the case of censored data. In spite of many similar parts of TNW-CATE and BENK, they are different because BENK is based on using the Beran estimator for training and can be successfully applied to survival analysis of controls and treatments. However, TNW–CATE and BENK use the same idea to train neural kernels: implementation as neural networks instead of using standard kernels.

It is also interesting to point out that BENK does not require oneto have a large dataset for training, even though the neural network is used for implementing the kernels. This is due to a special way that is proposed to train the network, which considers pairs of examples from the control group for training, as in Siamese neural networks. Our

experiments have illustrated the outperforming characteristics of BENK. At the same time, we have to point out some disadvantages of BENK. First, it has many tuning parameters, including parameters of the neural network and parameters of training $n$ and $N$, such that the training time may be significantly increased in comparison with other methods of solving the CATE problem. Second, BENK assumes that the feature vector domains are similar for controls and treatments. This does not mean that they have to totally coincide, but the corresponding difference in domains should not be very large. A method which could take into account a possible difference between the feature vector domains for controls and treatments can be regarded as a direction for further research. An idea behind the method is to combine the domain adaptation models and BENK.

Another direction for further research is to study robust versions of BENK when there are anomalous observations that may impact training the neural network. An idea behind the robust version is to use attention weights for feature vectors and also to introduce additional attention weights for predictions.

It should be noted that the Beran estimator is one of several estimators that are used in survival analysis. Moreover, we have studied only the difference in expected lifetimes as a definition of the CATE in the case of censored data. There are other definitions, for instance, the difference in SFs and the hazard ratio, which may lead to more interesting models. Therefore, BENK implementations and studies using other estimators and definitions of the CATE can be also considered as directions for further research.

The proposed method can be used in applications that are different from medicine. For example, it can be applied to selection and control of the most efficient regimes in the Internet of Things. This is also an interesting direction for further research.

**Author Contributions:** Conceptualization, S.K., L.U. and A.K.; methodology, L.U. and V.M.; software, S.K. and A.K.; validation, S.K., V.M. and A.K.; formal analysis, A.K. and L.U.; investigation, L.U., A.K. and V.M.; resources, A.K. and V.M.; data curation, S.K. and V.M.; writing—original draft preparation, L.U. and A.K.; writing—review and editing, S.K. and V.M.; visualization, A.K.; supervision, L.U.; project administration, V.M.; funding acquisition, V.M. All authors have read and agreed to the published version of the manuscript.

**Funding:** The research is partially funded by the Ministry of Science and Higher Education of the Russian Federation as part of World-Class Research Center program: Advanced Digital Technologies (contract No. 075-15-2022-311. dated 20 April 2022).

**Institutional Review Board Statement:** Not applicable.

**Informed Consent Statement:** Not applicable.

**Data Availability Statement:** Data are contained within the article.

**Conflicts of Interest:** The authors declare no conflicts of interest.

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
