# Peer review of "BENK: The Beran Estimator with Neural Kernels for Estimating the Heterogeneous Treatment Effect"

_algorithms, doi:10.3390/a17010040_

Round 1

Reviewer 1 Report

Comments and Suggestions for Authors

The paper describes the neural network model extended to beran estimator. Such hybridization is interesting, but some more details are needed. My main concerns are:

The abstract should be extended to more details about the novelty.

The authors used over 100 references, which most of the are outdated. Reduce this number to max 50. Therefore, i recommend rewriting the first two sections to the current state of art in machine learning.

Discuss the latest neural network models as: prediction model for smart homes via decentralized federated learning, neuro-heuristic analysis of video in iot system, etc.

Explain how exactly the controls' dataset is put into beran estimator.

What is the loss function?

How it can be implemented in some known framework like tensorflow? Some more details are needed.

More justification of the constructed network should be added.

The comparison with state of art is missing.

What about another kernel? Some discussion and comparison could be valuable for the paper.

Reviewer 2 Report

Comments and Suggestions for Authors

The text introduces a novel method named BENK (Beran Estimator with Neural Kernels) for estimating the conditional average treatment effect in the presence of censored time-to-event data. The approach utilizes the Beran estimator to estimate survival functions for both control and treatment groups. Unlike traditional kernel functions in the Beran estimator, BENK employs neural kernels, specific neural networks that enhance flexibility in modeling complex feature vector structures. The estimation of conditional average treatment effect is achieved by utilizing survival functions as outcomes from neural networks representing control and treatment groups, each comprising neural kernels with shared parameters. The method is evaluated through numerical simulation experiments, comparing its performance with T-learner, S-learner, and X-learner across various types of control and treatment outcome functions derived from Cox models, random survival forest, and the Beran estimator with Gaussian kernels. The text emphasizes the flexibility and accuracy of neural kernels in capturing intricate location structures within feature vectors. Furthermore, the code implementing BENK is made publicly available for wider accessibility. It is written in python.

The first four chapters present a very complete introduction to the methods of survival analysis, the need to define the "conditional average treatment effect" and the methods to estimate it. The main novelty of the paper is introduced in sections 5 and 6.

The point that I find most confusing is that to define the loss function, only the uncensored data are considered, while the Beran estimator uses all observations to estimate the survival function. Just out of curiosity, I would like to know if there is any known method that allows to define loss functions taking into account also the censored data.

The PEHE metric is used to represent the results of the  experiments.  In most cases, the results are shown graphically, and just the performance of a particular case is compared quantitatively  (Table 2). The only thing I would have liked would have been more commentaries on the results obtained.

As authors say, "one of the important peculiarities of the Beran estimator is that it takes into account distances between feature vectors by using kernels which measure how similar any two feature vectors are". So, I would have liked to have seen better how these simulated data look like. The authors detail the simulation process very well, but I miss some illustration of what the data samples used in the estimations look like. I am very struck by the fact that the feature spaces are the same for treatments and controls, and that the two sets only differ in the distributions of survival times.

(line 428) "In all experiments, feature vectors y are generated in the same way as vectors  x. However, for feature vectors x and y from the control and treatment groups, the corresponding times to events f and h are different and generated by using the Weibull  distribution".

Then, in the simulated data set, treatments and controls only differ in the distribution of the "times to event"? ¿Shouldn't they differ also in some characteristic of the feature space? Some how, authors are considering that between the treatments and the controls the baseline of the "patients" are exactly the same and there is only a treatment effect reflected in the survival times. Is that correct?

Could the method be applied if we had the feature vectors of the treatment group on the Archimedean spiral and the feature vectors of the control group on  two-dimensional circles, or this fact would distort the results?

Reviewer 3 Report

Comments and Suggestions for Authors

In this paper, the authors propose a machine-learning estimation technique that replaces the use of kernel functions with a neural network. This technique is used in the context of survival data, with an application to estimating the effect of heterogeneous treatment.

To estimate survival probabilities, the authors use the classic Beran estimator, with the difference that the Beran weights are estimated with a neural network (function K), which takes the (x,x_i)'s as input.
Although this idea is not completely original, as it seems that it has already been developed in [22], I find this case of application interesting.

I understand that the same sub-network is used to calculate K(x,xi_), K(x,x_j)... However, I'm wondering about the architecture of the sub-network used. I assume it's a classic feedforward network. How many hidden layers are there?

On the other hand, I wonder whether this network can really replace a kernel, in terms of the three properties normally expected of a kernel function, i.e. the non-negativity of the function, the normalization property (integral of the kernel function equal to 1) and the symmetry property. Without these properties, I wonder whether the Beran estimator is really consistent.

Is there a way to ensure these properties with the neural network?

Generally speaking, the asymptotic convergence of the proposed BENK estimator to the true survival function has not been proven.

The experimental results look good. I also like the fact that the code is shared, which makes it possible to replicate experiments. However, I have a few comments:

1) With the synthetic data used in this paper, the probability of delta_i being equal to 0 or 1 does not depend on (T,X), which is generally the case for survival data.

2) The probability of missing data is always 33%. What could be the impact for different values of the proportion of censored data ?

3) In Table 2, there are no statistical tests to support the results.

4) In Figures 4-12 errors bars are missing.

[22] Konstantinov, A.; Kirpichenko, S.; Utkin, L. Heterogeneous Treatment Effect with Trained Kernels of the Nadaraya–Watson 587
Regression. Algorithms 2023, 16, 226.

Round 2

Reviewer 1 Report

Comments and Suggestions for Authors

My comments were addressed.

Reviewer 3 Report

Comments and Suggestions for Authors

Thank you for the responses to my comments. Overall, I feel that these comments have been taken seriously in the new version.